# Possible impact of the 43 BCE Okmok volcanic eruption in Alaska on the climate of China as revealed in historical Document

Pao K. Wang[1,2,3], Elaine Kuan-Hui Lin[4], Yu-Shiuan Lin[1], Chung-Rui Lee[1], Ho-Jiunn Lin[1], Ching-Wen Chen[4] and Pi-Ling Pai[5]

[1]Research Center for Environmental Changes
Academia Sinica, Taipei, Taiwan
[2]Department of Atmospheric and Oceanic Sciences
University of Wisconsin-Madison, Wisconsin, USA
[3]Department of Atmospheric Science
National Taiwan University, Taipei, Taiwan
[4]Institute of Environmental Education
National Taiwan Normal University, Taipei, Taiwan
[5]Research Center for Humanities and Social Sciences
Academia Sinica, Taipei, Taiwan

*Corresponding to*: Pao K. Wang (pkwang@gate.sinica.edu.tw)

**Abstract.** The Okmok volcanic eruption in Alaska has been recently discovered and precisely dated to have occurred in 43 BCE. Some Chinese climate records of 43 - 33 BCE in historical documents have been found that provide descriptions of observed environmental abnormities that appear to be consistent with the anticipated changes due to volcanic climate forcing. We provide full translation with discussions of the Chinese climate records that may be related to the Okmok eruption in this paper. We have converted ancient Chinese calendar dates to modern Gregorian dates and provided the latitudes and longitudes of the geographical locations mentioned in the records. Some information about the few decades of post-Laki 1783 eruption climate condition in similar areas of China is also briefly summarized for comparison. We believe the detailed information contained in these records will be useful for further research on the climate impact of volcanic eruptions.

**Summary**

We provide detailed translation of some abnormal meteorological conditions in 43-33 BCE described in Chinese historical documents possibly related to the Okmok volcanic eruption in Alaska in early 43 BCE. The cold summer record and the abnormal color and low brightness of

the sun point to the clear link to the volcanic impact.  The reported duration for the visual
condition of the sun to return normal should be useful for researchers modeling the volcanic
impact on climate.

**1. Introduction**

It has been known for some time that volcanic eruptions are an important forcing in

shaping the global climate (Bradley 2015; Gao et al. 2008) and some recent events, such as
the eruption of Pinatubo in 1991 that caused discernable climate cooling have been studied
and reported (e.g., McCormick et al. 1995; Sukhodolov et al. 2018). Since climate change is
a globally urgent issue facing the human society and that predictions of future climate change
rely mainly on climate models which, at present generation, still produce results with large
uncertainties (IPCC, 2023), it is of great importance to improve and validate these models.
One common practice is to run these models to back-predict the past climate during a certain
period with known forcing terms and compare the model results with observations. An
example of such activities is PMIP which stands for Paleoclimate Model Intercomparison
Project (see, e.g., Jungclaus et al., 2017). But this requires high quality past climate data and
evidence of events that might indicate important climate forcing. Given high impact of
volcanic forcing on climate change, obtaining accurate volcanic eruption records is evidently
highly important.

Large explosive volcanic eruptions exert short-term cooling on the global climate which

counteracts the greenhouse gas-induced warming and, by doing so, may alter climate
conditions of certain regions. For example, this cooling has been suggested to reduce land-
sea thermal contrast and suppress summer precipitation, especially in low-latitude monsoon
regions (Gao and Gao, 2018; Iles & Hegerl, 2014; Schneider et al., 2009; Robock et al.,
2008). Changes in monsoon rainfall have great repercussions on the food production and
human societies in these areas. Thus, it is obviously important to understand these large
eruptions and their impacts on climate.

A recent study revealed a previously unreported volcanic eruption occurred in Mount

Okmok in Alaska with an unprecedented accurate dating technique and pinpointed that the
eruption occurred in early 43 BCE (McConnell et al. 2020). Such accurate dating is very
important in that it can link unambiguously with other records describing climate-related
phenomena observed at the same time to form a complete cause-and-effect chain, and such a
chain becomes a valuable data for climate model validation: Only those models that include
the right causes at the right moment through the right physical sequence and produce the
accurate effect as observed can be considered as validated for this forcing. The records
discussed here reveal such a cause-and-effect chain.
In interpreting the climate change, we also need to keep in mind that the climate is
governed by the complex interactions among various external forcing and internal modes,
and volcanic responses can invoke but also easily overridden by internal modes such as
ENSO. This is certainly true in East Asia also. Sometimes the climate signals of these factors
overlap and render the determination of the causes difficult. In addition, the reconstructed
climate change may look different when interpreted by different sets of proxy data. For
example, Gao et al. (2017) demonstrated that a discrepancy exists between the reconstruction
by Anchukaitis et al. (2010) based on the tree ring-derived Monsoon Asia Drought Atlas
(MADA, see Cook et al., 2010) and that by Chinese historical documents when analyzing the
climatic responses in China after the 1815 Tambora eruption. Later, Feng et al. (2013) used a
multiproxy-based reconstruction that supports the document-based reconstruction (Gao and
Gao, 2018). This underlines the importance of accurate data sets for climate studies.

**2. Okmok eruption in 43 BCE and contemporary Chinese climate records in 43-33 BCE**
Chinese historical documents contain many records that contain information about the
climate conditions of the time. Many of these have been utilized for the reconstruction of past
climate in China in the historical time (see, e.g., Wang, 1979, 1980; Wang & Zhang, 1988,
1991, 1992; Zhang & Wang, 1991). We have recently digitized the climate records in China
in the past 3000 years listed in Zhang (2013) by designing an extensive dictionary to convert
these records into digital form to build a climate database called REACHES such that
researchers can utilize these records even if they are not familiar with Chinese language
(Wang et al. 2018; Lin et al. 2020).
Among these ancient records, one that had caught our attention long time ago is the 'cold
summer' record dated at 43 BCE as it is the first such report with precise timing in an official

national chronicle, *Han Shu* (literally the History of Han Dynasty), about which more will be said later. This and other sequel records at that time are, in our opinion, of importance for understanding the impact of volcanic eruptions on global climate. They had been briefly mentioned in McConnell et al. (2020) but without much details. It is felt that by providing the full contents of these Chinese records, climate researchers can profit by digging deeper into this event and scrutinizing the meaning of the descriptions of the records. This will lead to a better understanding of the volcanic impact on climate both qualitatively and quantitatively.

In the following, we will provide full translations of these records that we deem relevant to the Okmok eruption along with our observations and interpretations that we believe would be useful.

We use the online utility http://www.nongli.net/sxwnl/ to convert the Chinese calendar to Gregorian calendar. The approximate latitudes and longitudes of the locations mentioned in these records were determined using the historical GIS developed in Academia Sinica (Liao and Fan 2012). If a record contains no specific location name, then it was an event usually observed at the national capital at the time, i.e., Changan (長安，34.03899° N, 108.9311° E). All events discussed below occurred during the reign of Emperor Yuan of Han Dynasty (漢元帝) who ruled China in the period 48-33 BCE. Starting in 140 BCE, it became a tradition of Chinese imperial systems to give a special name to the years of a certain period, called era name, during the reign of an emperor. There might be several such eras during the reign of an emperor if deemed necessary. Even though Emperor Yuan only reigned 16 years, he had four such eras: Chu Yuan (初元 48-44 BCE), Yong Guang (永光 43-38 BCE), Jian Zhao (建昭 38-33 BCE), and Jing Ning (竟寧 33 BCE).

All records discussed below were derived from the following five original Chinese historical documents as well as in Zhang (2013):

#1 – Annals of Emperor Yuan, *Han Shu* (漢書 元帝紀)

#2 – Records of Five Elements, *Han Shu* (漢書 五行志)

#3 – Biography of Feng Fengshi, *Han Shu* (漢書 馮奉世傳)

#4 – *Lord Fu's Notes of Ancient and Contemporary Affairs* (伏侯古今注)
#5 – *Comprehensive Reflections to Aid in Governance* (資治通鑑)
The first three documents are all from *Han Shu* authored by Ban Gu (32-92 AD) who was
the pioneer of Chinese chronological history. #2 contains a large amount of observed
abnormal environmental phenomena. #4 was written by Fu Wuji (circa 130 AD). Both Ban
Gu and Fu Wuji lived in Han Dynasty. #5 was a comprehensive reference compiled in Song
Dynasty led by Sima Guang (1019-1086 AD) based on the imperial historical documents. We
use the numerical indices to indicate the source of the records (at the end in parenthesis) in the
following discussions. The records are listed in chronological order.
Fig. 1 shows a map of the locations mentioned in the discussions below.

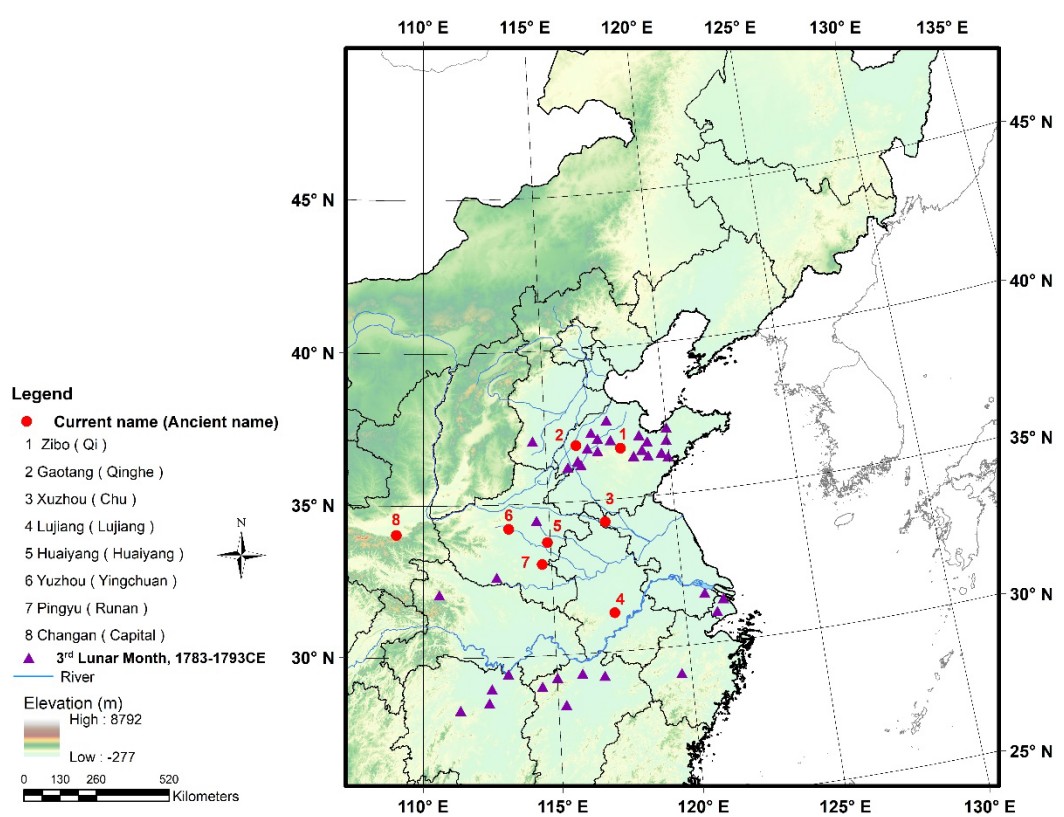


Fig. 1. A map showing the locations mentioned in the text. Red circular dots are those
associated with the Han dynasty records (location names in the legend). Purple triangles are
those associated with the cold records in the general area of central China in the period of

137 1783-1793.

**a) 43 BCE (Yong Guang 1ˢᵗ year)**

    i) "In 3ʳᵈ Month (8 April – 6 May), snowfall. Frost damaged wheat crops. No harvest in the fall" (#1)

    ii) "In 3ʳᵈ Month, frost damaged mulberry" (#2)

    iii) "In 4ᵗʰ Month (7 May – 5 June), the sun was bluish-white in color and casted no shadow. When the sun reached the zenith, it did cast shadow but had no glare. The summer was cold. The glare of the sun recovered in the 9ᵗʰ Month (2 - 31 October)" (#2)

    iv) "On 2ⁿᵈ Day of 9ᵗʰ Month, frost damaged crops. Severe famine occurred in the whole country" (#2)

Of the above four records, iii) is the most directly relevant to the volcanic eruption, hence we will discuss it first. Coloration of the sky and that of the celestial objects in it can be an important indication of the presence and the altitude level of the volcanic dusts. For example, Guillet et al. (2023) utilized the coloration of the moon during total moon eclipses to determine the stratospheric turbidity so as to infer the occurrence of climate-forcing volcanic eruptions. In this record, the original Chinese characters describing the color of the sun in this record was 青白 (qing bai) which can be translated as "greenish-white" or "bluish-white" due to the somewhat ambiguity of the meaning of "qing" in ancient Chinese language, as it could mean either "bluish" or "greenish", but we shall use bluish-white for our discussion here. The description of sun color here already indicated that it was unusual, and the most likely cause, in light of the discovery of Okmok eruption, was that the sun was veiled by a thin layer of volcanic dusts in the sky. Such blue sun (and moon) phenomenon caused by volcanic ash has been observed repeatedly and lasted hours for a time during the 1883 Krakatau eruption (Minnaert, 1993).

The second important indication of the presence of volcanic ash is that the sun casted no shadow except when it was at zenith. Again, this was likely due to the presence of the volcanic dusts that scattered sunlight, rendering the sky light a diffuse light source which therefore casted no shadow (Minnaert, 1993). This effect is more pronounced when the sun angle is low in the morning or in the later afternoon as the sun's rays have to go through a thick layer of the atmosphere. When the sun is at the zenith, its rays go through a much

thinner layer of the atmosphere and therefore suffers less scattering and is capable of casting a
shadow. But obviously the scattering was substantial enough to reduce the glare of the sun as
described by the record.
The record indicates that that summer was cold. The use of 'cold' (寒) to describe summer
condition was rather unusual in Chinese historical records and must indicate a rather severe
departure from the norm. A few cold days in a summer may be not so unusual, but a whole
cold summer season must be extremely rare. Thus, we feel that the estimate of 2°C colder
than normal mean given in Tan et al (2003) is reasonable. A cooling of such a magnitude was
possible under the strong volcanic radiative forcing associated with the Okmok eruption.
The record then says that the sun glare recovered in the 9th Month, roughly 5 months after
the sighting of the unusual sun color. This should indicate how long the volcanic dust hovered
over northern China in 43 BCE. This information should be of importance to researchers
trying to model the cooling and those interested in modeling the transport of volcanic dust to
China from Okmok.
Now we can go back to examine records i) and ii), both indicate cold condition in the 3rd
month. Even though these events occurred before the sighting of the volcanic dusts, it was
still possible that the cold climate was caused by the volcanic forcing as the Spring time
weather of northern China is usually influenced strongly by the movements of polar air
masses. Okmok is located much further north than China, and the cold air mass originated in
Alaskan polar region can certainly influence the spring weather in Northern China. The
volcanic forcing could have caused colder-than-normal air masses that resulted the frosty 3rd
Month in China when they moved south.
Record iv) can be interpreted in a similar way. Even if the volcanic dusts had disappeared,
it is still possible that forcing effect lasted longer and hence the frost and famine could still be
attributed to the volcanic event.
**b)** After 43 BCE
It is known that the impact of volcanic eruption on climate can last many years if the dusts
reach high in the stratospheric level such as the case of Pinatubo eruption in June 1991
(McCormick et al. 1995). Hence it is also useful to list relevant climate records a few years
after the eruption event. In the next section, we list those records within 10 years after the 43
BCE Okmok eruption.

**i).  42 BCE (Yong Guang 2$^{nd}$ year)**

• In 6$^{th}$ Month (24 July – 22 August), the imperial decree declared "Recently, there
are years of poor harvest and all areas are in serious condition. People worked hard
on tilling but received no produce. They are suffering from famine and there is no
relief" (#1)
• At this time, there were many crop failures, …, all areas are suffering famine" (#3)
These two records are essentially saying the same thing, namely, poor crop yield led to
famine which could be attributed to the cold climate. However, the term 'years' could
mean two or more years and therefore the climate that resulted in famine might or might
not relate to the Okmok eruption.

**ii)  41 – 40 BCE (Yong Guang 3$^{rd}$ year)**

• In 11$^{th}$ Month (7 December, 41 – 5 January, 40 BCE), the imperial decree declared
"(It) rained in mid-winter and heavy fog (occurred)" (#1)
The words in parenthesis are added by us to render the sentence easier to understand in
English. Rain in mid-winter is extremely rare in northern China now as well as then where
the capital of Han Empire, Changan, is located and this statement must indicate a severe
anomaly. Rain occurred in midwinter was presumably because the unusual warm weather
at this time. This was obviously not directly due to the negative radiative forcing of the
volcanic dusts, but could it be a climatic repercussion of the severe coldness of the
previous year? Similarly, the fog must have been extraordinary heavy to deserve a
mention in the decree. In addition, fog consists of liquid droplets (since the statement did
not say ice fog) and therefore this also indicated abnormally warm climate that winter. It
is not known that fog can be directly related to a volcanic event but it could also be a
result of repercussion. Both require further study in the future.

**iii) 39 BCE (Yong Guang 5$^{th}$ year)**

• In the fall (7 August – 6 November), Yingchuan (潁川 34.19589N, 113.3792E)
flooded and killed people (#1)

- Heavy flood in summer (5 May - 6 August) and fall. Rain in Yingchuan, Runan (汝南 32.99044°N, 114.6317°E), Huaiyang (淮陽 33.70539°N, 114.8841°E) and Lujiang (廬江 31.26964°N, 117.3212°E) damaged houses in rural areas and causing flood that killed people. (#2)
- In this year, the Yellow River flooded at Lingmingdu Mouth （靈鳴犢口） of Qinghe (清河 36.83046°N, 116.2479°E), but River Tunshi (屯溪 a tribute of Yellow River) dried out. (Vol. 21, History of Han, #5)

All of these records mentioned flood and the second entry seems to indicate that the flood was caused by heavy rain. Again, they were not directly related to the volcanic eruption but might be its climatic repercussion.

**iv) 38 BCE (Jian Zhao 1st year)**

- In 8th Month (7 September – 6 October), large swarm of flying white moths shrouded the sun (#1)

This is also not directly linked to the volcanic event but it is also possible that the unusual biospheric phenomena might have been caused by the abnormal climate condition due to the repercussion.

**v) 37 – 36 BCE (Jian Zhao 2nd year)**

- In 11th Month (23 December, 37 – 20 January, 36 BCE), earthquake occurred in Qi and Chu. Big blizzard broke trees and damaged houses (#1)

The earthquake should not be related to the Okmok eruption, but the cold climate that led to the strong blizzard could be due to it.

- In 11th Month, big blizzard occurred in Qi (齊 36.64394°N, 118.0556°E) and Chu (楚 34.27161°N, 117.2056°E) areas and was 5 *chi* （尺） deep (#2)

The information in this record is essentially the same as he one above but it gave an additional information on the snowfall amount, 5 *chi*. *Chi* is a Chinese length unit whose length varied from time to time historically. There were Han rulers unearthed and it was determined that one *chi* in Han dynasty is roughly 23.1 – 23.3 cm (Hsu 2009). 5 *chi* is

therefore roughly 116 cm or 46.4 in, certainly an unusually heavy blizzard in these
locations that could cause the disasters reported in the previous record.
• Jing Fang (77-37 BCE) from Dong Jun spoke to Emperor Yuan about the
disasters and abnormities, "Ever since Your Majesty ascended the throne, the sun and
the moon had lost their glares, stars orbited reversely, mountains collapsed and springs
gushed out from underground, the earth quaked and rocks fell, frost appeared in
summer and thunders heard in winter, plants withered in spring and flowered in fall,
frost unable to kill plants, and flood/drought and locust outbreaks occurred. People
suffer from famine and plagues, bandits cannot be suppressed, and prisoners are
everywhere. All the disasters and abnormities mentioned in *Chun Chiu* (a chronicle of
Lu Dukedom edited by Confucius) have happened" (Vol. 21, History of Han, #5)
According to traditional Chinese belief, abnormal natural phenomena, be it
astronomical or earth environmental, occur because they reflect the health state of the
political system. When auspicious phenomena (such as colorful clouds or large group
gathering of cranes) occur, it must indicate that the system is running well and the
emperor was considered virtuous and fit to rule. If ominous signs (such as what
mentioned in this records) occur, then there must be something wrong in the system, and
ideally a faithful government official should not be afraid to tell the truth to the emperor.
These uncomplimentary comments from Jing Fang, a procurator and scholar known for
his studies in divination, must have been very unpleasant to the royal ears as he attributed
all these disasters and abnormities to the incompetent rule of Emperor Yuan. It took a
great courage for a low-level official to take such an action but this also indicates that
what he said about the abnormal climate events must have occurred, for otherwise it
would be purely suicidal to make such statements.
Unfortunately, Jing Fang was framed by the head eunuch, Shi Xian, whom was the
real target of Jing Fang's attribution, and eventually died in jail. Attributing these climate
abnormities to political incompetence is obviously unscientific, but there is no way Jing
Fang could have known that the real culprit was a volcano some 6000 km from his
country!
**vi) 35 BCE (Jian Zhao 4th year)**
• Dustfall (#4)
Unfortunately, there is no precise month given in this record and it was unclear whether this
had a connection with the volcanic eruption or not.
There is another record listed under this year stating that "In 3rd Month, snowfall occurred
and many swallow died". However, this is possibly an error and the event should belong to one
in 29 BCE, and the month should be 4th Month (Shi, 1994). This is beyond the 10-year period of
interest here and will not be discussed.
**vii) 33 BCE (Jing Ning 1st year)**
• Heavy fog. All trees turned white. (#4)
Like the previous record, this record does not contain the month information and we don't
know which season it belonged. It is also unknown why trees turned white. However, one
possibility of trees turning white is that this was a freezing fog event such that fog droplets stuck
on trees and turned into ice. If so, then this record can possibly be interpreted as indicating a
colder-than-usual condition, especially if the fog did not happen in winter.
**3. A brief comparison with the possible responses in China in post-Laki period**
As mentioned before, the climate is governed by the complex interaction of many factors,
therefore what described in the previous section should be taken as possible, but not definite,
climate response of the Okmok eruption in China. Nevertheless, we believe the possibility is
high, as we observe a similar climate fluctuation pattern in central China after another large
eruption in 1783-1784, the Laki eruption in Iceland. Like Okmok case, the Laki eruption was a
high-latitude event and a strong one with a VEI (volcanic eruption index) at 4, and therefore we
would expect that it would have impact on the climate fluctuation in China at that period. Since
the winter season is normally cold in central and northern China, it would be more difficult to
attribute cold winters to the influence of volcanic eruption. Instead, we show the evidence of
cold climate in the 3rd month which corresponds to late spring in Chinese lunar calendar. This
season was generally regarded as warm and a time for flowers to blossom. Frost or snow in this
season should then indicate colder than normal condition. Fig. 2 shows the frequency of frost
and/or snow records in the 3rd month of the period 1733-1833 in northern and central China as
those mentioned in the last section. The exact locations of the records associated with the post-
Laki decade (1783-1793) are shown in Fig. 1 as purple triangles. We can see that they overlap
generally with those locations mentioned in Section 2.

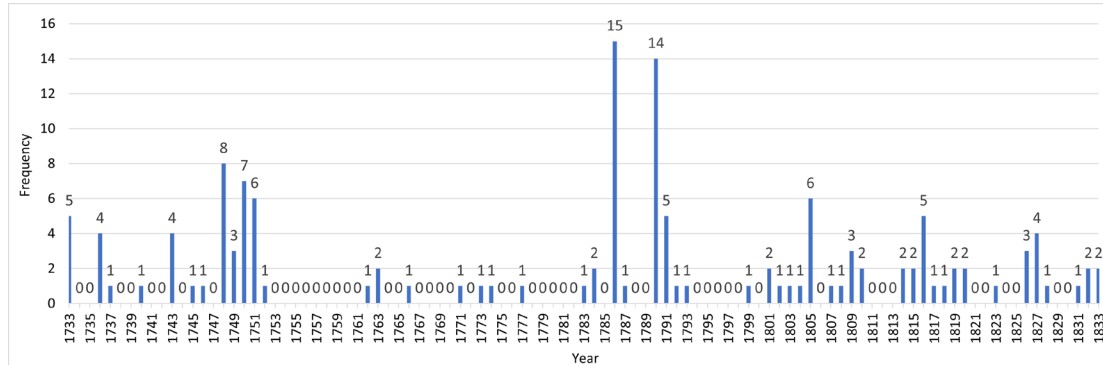


Fig. 2. Annual number of cold condition records in northern and central China in the period
of 1733-1833 derived from REACHES database.
Fig. 2 shows that the annual number of cold records in northern and central China had a high
period in 1748-1751 which, if due to volcanic factor, could be associated with the Oshima-
Oshima eruption in Japan in 1741-1742 (Smithsonian Institute, 2013) but the impact would occur
many years later. The cold condition then subsided considerably for the next 35 years. Then two
very high peaks in 1786 and 1790 and a moderate peak at 1791 occurred, several years after the
Laki eruption.  After that, a series of peaks occurred at 1805, 1816, and 1827, a roughly 11-year
periodicity. The timing of the 1786-1791 peaks suggests that they could be due to the volcanic
radiative forcing of Laki eruption as the impact can occur a few years after the eruption. On the
other hand, it is less certain what were responsible for the cold peaks in the 19th century.

## 4.  Conclusions

In the above, we translated several climatic records kept in Chinese historical chronicles
for the 10-year period (43 – 33 BCE) after the Okmok eruption at 43 BCE recently identified
(McConnell et al., 2020). These records clearly portrait a generally cold and harsh climate period
that was commensurate with the negative radiative forcing expected for a volcanic eruption.
Descriptions of the observed optical abnormities of the sun and moon also match the expected
consequences due to the veiling of high-altitude volcanic dusts, and the veiling might have lasted
as long as 6 months. Such a long veiling period at such a long distance away from the source
should indicate that the eruption must be of extraordinary magnitude as suggested in McConnell
et al. (2020).
The precise dating of volcanic eruptions such as the studies in Gao et al. (2008) and
McConnell et al. (2020) is obviously very important for identifying the cause or forcing
responsible for certain past climate conditions such as the cold summer of 43 BCE recorded in
the Chinese history which otherwise would always remain as a mystery. Conversely, there are
many other similar climate records listed in Chinese historical documents that can be used for
reconstructing past climates and their environmental impact, and when combined with new
technologies such as done in Gao et al. (2008) can significantly advance our knowledge about
the science of climate change (Wang et al. 2018; Lin et al. 2020).
**Acknowledgments.** We thank Dr. C. C. Gao and Dr. Philip Gooding for their constructive
suggestions that lead to the improvement of the original manuscript. Editorial help from Dr.
Alberto Reyes is also highly appreciated. This work is partially supported by National Sciences
and Technology Council of Taiwan grant NSTC 112-2122-M-001-001.

**Code/Data Availability.** No code or new data is used in this work.
**Author contribution.**
Pao K. Wang – project holder, perceived the paper and wrote the original draft
Elaine Kuan-Hui Lin – reviewed and edited the draft
Yu-Shiuan Lin – data extraction and GIS operation
Chung-Rui Lee – reviewed and edited the draft and consistency check
Ho-Jiunn Lin - reviewed and edited the draft
Ching-Wen Chen - reviewed and edited the draft
Pi-Ling Pai - reviewed and edited the draft
**Competing interests** – None

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
