# Peer review of "Possible impact of the 43 BCE Okmok volcanic eruption in Alaska on the climate of China as revealed in historical Document"

_EGUsphere, 2024_

## Author Response (AR1)

Review comments and response (in red)

Dr. Chaochao Gao

The paper entitled "Possible impact of the 43 BCE Okmok volcanic eruption in Alaska on the climate of China as revealed in historical documents" compiles the Chinese documentary evidence of the potential climatic consequence following the Okmok eruption, which provides valuable complimentary for the McConnell et al. (2020) study and paleo-reconstruction check for future modeling investigations. I therefore recommend publication of the study after addressing the following issues:

1. Due to the reducing documentary records as we go back in time, it would be helpful to compare and discuss the extreme climate during the years surrounding the Okmok within the long-term context of the last 2000-yr, especially the latest few centuries where the records are abundant. For example, how often is "snowfall and frost damage" during the 3rd month in more recent centuries in central China? How do the climatic consequences compare to those during the post-Laki years?

   We have added a new Sec. 3 to discuss the post-Laki climate condition of northern and central China to show that indeed similar cold conditions occurred in that period. A new figure (Fig. 2) is added to show the change. The relevant locations mentioned in this section are also marked in Fig. 1 (also a new figure itself).

2. One significant feature of the REACHES reconstruction (Wang et al., 2018) is the digitalization of the specific geographic location of the recorded events. Therefore, it would be nice to draw a figure illustrating the locations of the relevant climate consequences where the information is available.

   Fig. 1 and Fig. 2 are added for this purpose.

3. Numerous studies have been done on the spatial-temporal patterns of climate responses to volcanic eruptions, including Northern Hemispheric ones. A brief discussion of these findings, and how do they associate with the records synthesize in this study would be beneficial.

   A brief discussion of these is added in Sec. 1.

4. It is important to also mention that climate in East Asia is governed by the complex interactions among various external forcing and internal modes, and volcanic responses can invoke but also easily overwritten by internal modes such as ENSO.

   This is also added in Sec. 1.

5. Please add Guillet et al. (2023, cited below) and similar references on the direct observation of the volcanic eruptions' radiative appearance to the discussion of Records # iii in 43BCE, since it is one of, if not the most critical evidence of the Okmok eruption's influence in China.

   Added.

References:

Guillet, S., Corona, C., Oppenheimer, C. et al. Lunar eclipses illuminate timing and climate impact of medieval volcanism. Nature 616, 90–95. https://doi.org/10.1038/s41586-023-05751-z, 2023

McConnell, J.R., Sigl, M, Plunkett, G., et al: Extreme climate after massive eruption of Alaska's Okmok volcano in 43 BCE and effects on the late Roman Republic and Ptolemaic Kingdom, Proc. Natl. Acad. Sci. U.S.A., 117, 15443–15449, doi:10.1073/pnas.2002722117, 2020.

Wang P. K., Lin, K. E., Liao, Y. C., et al.: Construction of the REACHES climate database based on historical documents of China, Nature: Sci.Data, 8, 180288, doi:10.1038/sdata.2018.288, 2018.

Dr. Philip Gooding

"Possible impact of the 43 BCE Okmok volcanic eruption in Alaska on the climate of China as revealed in historical documents" expands on existing studies of the Okmok eruption in two core ways: i) by adding evidence from China; and ii) by incorporating evidence from documentary materials (the "archives of society"). I agree with all the comments made by the first reviewer. In addition:

1. Can the authors give a sense of how unusual the documentary reports on climate in China were in the years covered in the piece compared to those of in the circa 5 decades either side of the 43 BCE eruption? E.g. Was frost in the 3rd month common in the 1st century BCE, according to the records? I note that the first reviewer requested a comparison with more recent years, including after more recent eruptions (Laki, 1783). But, if the records allow, a comparison with more contemporaneous years would be useful as well. If it is not possible, then an explanation about the limitations of the sources would be beneficial.

   This is done in Sec. 3 with new figures (both Fig. 1 and 2).

2. There are a number of small grammatical errors. I hope that the journal editors will work with the authors to eliminate these before publication.

   We have checked and hopefully no typos in the revision.

I recommend that the piece be published after the requests for revision have been addressed.

---

## Author Response (AR2)

Reply to review comments

3: "documents" not "Documents", in this context
Corrected

20: "A massive eruption of Okmok volcanic eruption in Alaska…" is awkward. Suggest"The Okmok volcanic eruption in Alaska…"or "A large eruption of Okmok volcano in Alaska…"
Changed to "The Okmok volcanic eruption in Alaska…"

74: "but also are easily overwritten …'?
Changed to "overridden"

Fig 1: Caption (and text at line 312) mentions blue triangles but the triangles in the figure are purple. Also, the legend and caption text associated with the triangle symbols appears to be different re: the date range. What does the green shading indicate? Lettering for the lat/long graticule is strangely faded and should be fixed for
the final version.
Changed "blue triangles" to "purple triangles" (also corrected in Line 312). The date range in the caption text is changed to 1783-1793. The date range in the text in Section 3 (line 312) is also corrected. The green shading indicates the elevation (added in the legend). The font of lat/long graticule is changed.

175-177: This statement is stronger in attributing the climate phenomenon to Okmok, compared to the previous manuscript version. Not sure if this was intended? The abstract and summary are less certain in terms of the attribution.
I have toned it down now.

179: "dust" not "dusts"
Corrected.

313/314: Suggest "Section 2" instead of "last section", to reduce potential confusion
Changed.

Acknowledgements: If using the PhD title for one referee might as well use it for both.

Corrected. Also, thanks for your help (added also in the acknowledgments).